# Assembly and substrate recognition of curli biogenesis system

Zhaofeng Yan [1,2,3,5], Meng Yin[1,2,3,4,5], Jianan Chen[1,2,3,5] & Xueming Li [1,2,3,4]*

A major component of bacterial biofilms is curli amyloid fibrils secreted by the curli biogenesis system. Understanding the curli biogenesis mechanism is critical for developing therapeutic agents for biofilm-related infections. Here we report a systematic study of the curli biogenesis system, highlighted by structural, biochemical and functional analysis of the secretion channel complexes (CsgF-CsgG) with and without the curli substrate. The dual-pore architecture of the CsgF-CsgG complex was observed and used to develop an approach to inhibit the curli secretion by physically reducing the size of the CsgF pore. We further elucidated the assembly of the CsgFG complex with curli components (CsgA and CsgB) and curli-cell association through CsgF. Importantly, the recognition of the CsgA substrate by CsgG was uncovered. Nine crevices outside of the CsgG channel provide specific and highly-conserved recognition sites for CsgA N-terminus. Together with analysis of CsgE, our study provides comprehensive insights into curli biogenesis.

[1] Key Laboratory of Protein Sciences (Tsinghua University), Ministry of Education, Beijing, China. [2] School of Life Sciences, Tsinghua University, Beijing, China. [3] Advanced Innovation Center for Structural Biology, Tsinghua University, Beijing, China. [4] Tsinghua-Peking Joint Center for Life Sciences, Beijing, China. [5] These authors contributed equally: Zhaofeng Yan, Meng Yin, Jianan Chen. *email: lixueming@tsinghua.edu.cn

Bacteria have developed several highly specialized secretion systems (type I–IX) to adapt to the environment by secreting a wide range of substrates through the cell envelope into the extracellular milieu or host cells[1]. The curli biogenesis system in *Enterobacteriaceae* such as *Salmonella* species and *Escherichia coli* (*E.coli*), also known as the nucleation-precipitation or type VIII secretion system, is responsible for secreting curli subunits into the extracellular milieu to form cell associated nonbranching and highly aggregative amyloid fibrils[2,3]. Unlike the pathogenic amyloids involved in human degenerative diseases, curli amyloids promote biofilm formation to protect bacteria from physical and chemical stresses, host immune responses and therapeutic antimicrobial agents[4–9]. Bacterial biofilms play a crucial role in many infectious diseases, such as indwelling venous catheter sepsis, prosthetic-valve infective endocarditis, and Foley catheter-associated urinary tract infections (CAUTIs)[4,9–11]. Therefore, understanding the curli biogenesis mechanism is critical for developing therapeutic antimicrobial drugs for biofilm-related infections[12,13].

In *E. coli*, two separate operons (*csgBAC* and *csgDEFG*) produce seven proteins (CsgA, CsgB, CsgC, CsgD, CsgE, CsgF, and CsgG) that cooperate to form the curli amyloids[2,14]. CsgA is the major curli subunit, and structurally constitutes the amyloids fibers[9,15]. CsgB is a nucleator for nucleating CsgA subunits into amyloid fibers and assists the fibers to bind on the cell surface[16,17]. The architectures of two curli subunits, CsgA and CsgB, are similar, containing three domains, a signal peptide ($CsgA_{SP}$ and $CsgB_{SP}$), an N-terminal segment ($CsgA_{N22}$ and $CsgB_{N23}$) and a C-terminal amyloid core domain consisting of five repeating units ($CsgA_{R1-5}$ and $CsgB_{R1-5}$)[9,15]. CsgC localizes in the periplasm, and is thought to act as an inhibitor to prevent pre-amyloid formation and toxicity to the cell[18]. The expression of *csgBAC* operon is controlled by the transcriptional regulatory protein CsgD from *CsgDEFG* operon[19]. Lipoprotein CsgG forms a nonameric secretion channel in the outer membrane and transports CsgA and CsgB from periplasm into extracellular environment[3,20–22]. Two soluble accessory factors, CsgE and CsgF, have been thought to bind to CsgG and play a crucial role in substrate (CsgA and CsgB) secretion and curli amyloid fiber formation[21–24]. As the importance of the structures involved in secretion through the Csg complex channel, efforts have been made for determining Csg protein structures. The outer membrane channel CsgG complex as determined by X-ray crystallography contaians a large nonameric channel with a narrow inner pore[22]. Further cryoEM study for CsgG-CsgE complex showed that CsgE can bind at the CsgG channel periplasmic end[22]. The structures of individual Csg proteins, including CsgF and CsgE[13,25], has been studied by the nuclear magnetic resonance (NMR), and the crystal structure of CsgC has also been determined[26]. However, protein component assembly, explicit roles of CsgE and CsgF, and substrate recognition and transportation mechanisms are still not clear.

To address these questions, we determined the structures of complexes in curli biogenesis system by electron cryo-microscopy (cryoEM), including CsgF and CsgG (CsgFG) complexes both in the free for and bound to the N terminus (residues 1–22) of CsgA ($CsgA_{N22}$). Based on the two structures and corresponding biochemical and functional analyses, we systemically investigate curli biogenesis system assembly and CsgA recognition by CsgG channel, and discuss CsgE role in enhancing secretion efficiency. In addition, a feasible way to inhibit the curli secretion was proposed and examined by a specifically designed peptide inhibitor.

## Results

**CsgFG complex structure and CsgF-CsgG interaction**. The *E. coli* K12 *CsgEFG* operon was cloned and the proteins were

overexpressed and purified. The size-exclusion chromatography and SDS-PAGE confirmed that CsgF and CsgG can form a stable complex (Supplementary Fig. 1a, b). Despite co-expression with CsgFG, CsgE was not detected in the purified complex by Coomassie blue staining (Supplementary Fig. 1b), which may be related to the dynamic binding with CsgG in vivo. The CsgFG complex structure was determined by single-particle cryo-EM at 3.8 Å resolution without any imposed symmetry (Supplementary Fig. 1c–e). CsgG and CsgF were observed to bind together at a 9:9 stoichiometric ratio (Supplementary Fig. 1f). By imposing C9 symmetry during three-dimensional (3D) reconstruction, the resolution was improved to 3.38 Å (Fig. 1a and Supplementary Fig. 1e), which enabled unambiguous model building of entire CsgG subunit and N-terminal domain of CsgF ($CsgF_N$, residues 1–36, Fig. 1b and Supplementary Fig. 2). In the 2D class averages, the CsgF C-terminal domain ($CsgF_C$, residues 37–119) showed weak densities on the extracellular side of the channel (Fig. 1c and Supplementary Fig. 1d), which reflected high flexibility of this domain in the complex. Structural alignment with root-mean-square deviation (rmsd) of 0.742 Å showed that CsgG subcomplex in the CsgFG complex was similar to the individual CsgG nonamer at 3.59 Å resolution reported in a previous study[22] (Supplementary Fig. 3a), indicating that binding CsgF does not influence the CsgG channel structure.

$CsgF_N$ was inserted into the CsgG channel β-barrel to form a layer inside the CsgG channel (Fig. 1a, b). In addition, an isothermal titration calorimetric (ITC) assay showed that the dissociation constant of the CsgF-CsgG complex was about 0.289 μM (Supplementary Fig. 3c). Such strong interactions created a tight bond between CsgF and CsgG channel to provide stable curli-cell association through CsgF. Inside the CsgG channel, $CsgF_N$ was properly folded and contained a short α-helix (Fig. 1b). A previous NMR study reported that the $CsgF_N$ monomer was unfolded in solution[25] (Supplementary Fig. 4a). We examined this difference using circular dichroism (CD) and detected the α-

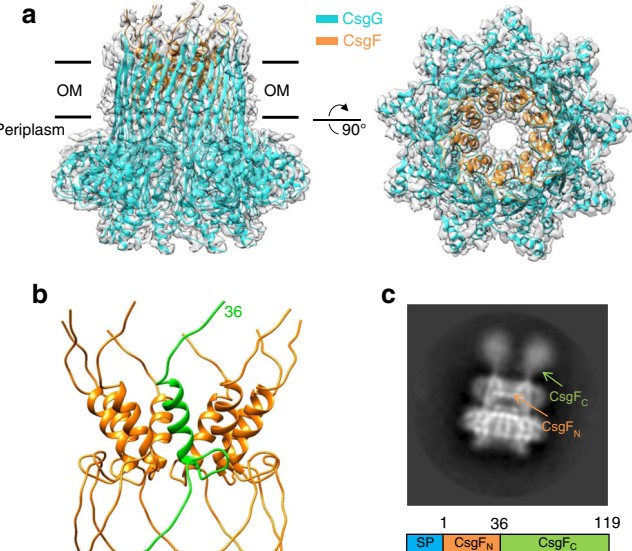

**Fig. 1 Overall structure of the CsgFG complex. a** Side (left) and top (right) views of the atomic model of the CsgFG complex superimposed with the corresponding cryoEM density map. Models of CsgG and CsgF are in cyan and orange, respectively. OM, outer membrane. **b** The nonameric $CsgF_N$ separated from the CsgFG complex. One of the $CsgF_N$ protomers is shown in green. **c** Representative 2D classification of the CsgFG complex. $CsgF_N$ and $CsgF_C$ are pointed by arrows. The schematic diagram of full length CsgF is shown at the bottom. SP signal peptide.

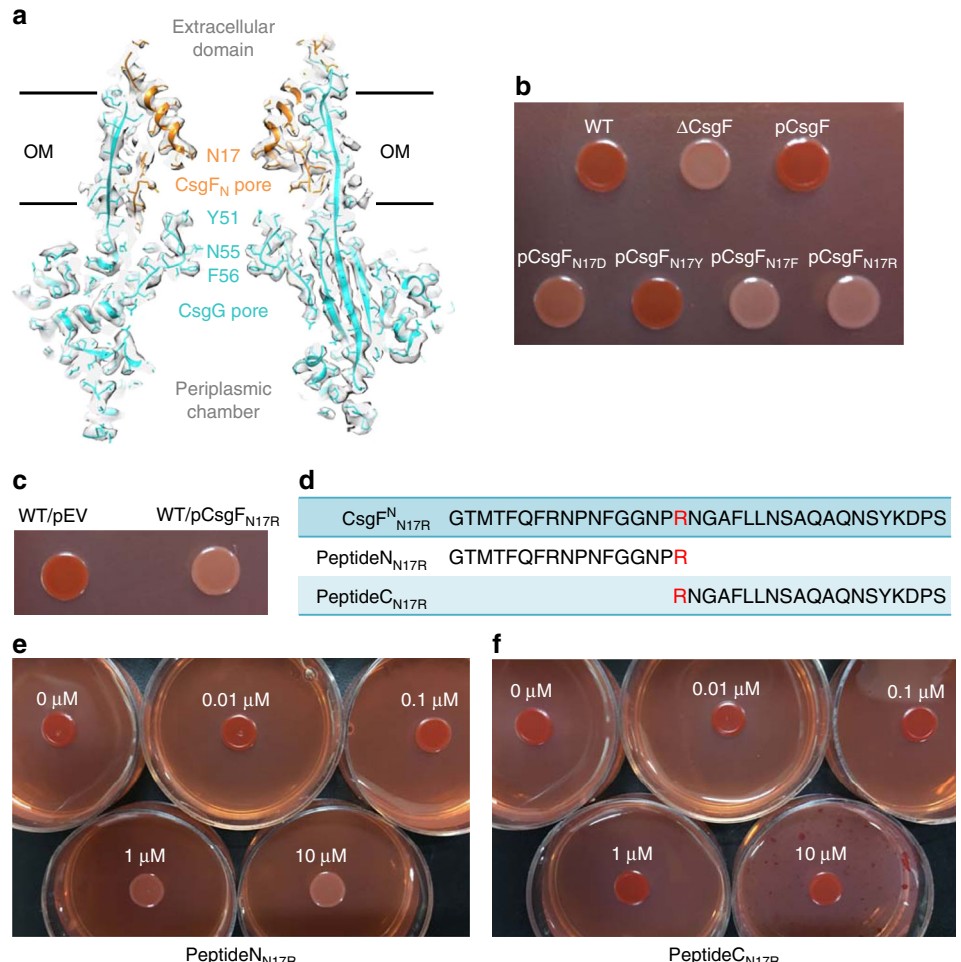

**Fig. 2 Dual-pore structure of CsgFG complex and analysis of peptide inhibition. a** Section view of CsgFG complex. The atomic model was superimposed with the corresponding cryoEM density map. Models of CsgG and CsgF are in cyan and orange, respectively. **b** Congo red secretion assay for mutations of the asparagine residue of CsgF$_N$ into four different kinds of amino acids in the ΔCsgF strain. Source data are provided as a Source Data file. **c** Congo red secretion assay for the N17R-mutated CsgF in the wild type (WT) strain. Source data are provided as a Source Data file. **d** Schematic diagram of the synthesized peptides, PeptideN$_{N17R}$ and PeptideC$_{N17R}$. The N17R mutation is shown in red. **e** Congo red secretion assay for the WT strain treated with PeptideN$_{N17R}$ at concentrations ranging from 0.01 to 10 μM. Source data are provided as a Source Data file. **f** Congo red secretion assay for the WT strain treated with PeptideC$_{N17R}$ at concentrations ranging from 0.01 to 10 μM. Source data are provided as a Source Data file.

helix formation in CsgF$_N$ after binding to CsgG (Supplementary Fig. 4b), indicating that CsgF$_N$ changed from the unfolded state to the folded state after binding to the CsgG channel.

Each CsgF$_N$ subunit binds to two adjacent CsgG subunits mainly by hydrogen-bonds, electrostatic interactions, and hydrophobic interactions (Supplementary Fig. 5a–d). To identify the key residues responsible for CsgF$_N$-CsgG interactions, we used CsgG as bait to pulldown CsgF mutant variants (Supplementary Fig. 5e). While two mutations, N11A and F21D, did not show any detectable influence on CsgF-CsgG interaction, four mutations, R8A, N9A, L22D, and L23D, showed reduced interaction between CsgF and CsgG (Supplementary Fig. 5e). However, Congo red cell-association phenotypes of these four mutations were similar to wild-type phenotype (Supplemenatry Fig. 5f), indicating that the ability of curli fibers to bind the cell surface is not significantly influenced by these reduced interactions. Three phenylalanine mutations (F5D, F7D, and F12D) abolished the interaction between CsgF and CsgG (Supplementary Fig. 5e). The Congo red cell-association phenotypes of these three phenylalanine mutants were nearly eliminated, confirming that these mutations destroyed the CsgF-CsgG interaction (Supplementary Fig. 5f). Considering that

Therefore, these phenylalanine residues are more critical for CsgFG complex formation than others in CsgF$_N$.

**CsgF pore and a peptide inhibitor.** Tyr51, Asn55, and Phe56 residues of CsgG were reported to constitute three stacked concentric rings to form a pore structure in the nonameric channel, which was essential to limit the passage of the folded proteins[22] (Fig. 2a). In the presence of the CsgFG complex, we observed another pore structure formed by Asn17 of CsgF$_N$ (Fig. 2a and Supplementary Fig. 6a–c), which was slightly larger (diameter of ~10.3 Å) than the one formed by CsgG (diameter of ~9 Å) (Supplementary Fig. 6b, c). The CsgF$_N$ pore may strengthen the restriction for the passage of the folded proteins, which might explain the considerably reduced leakage of non-native client proteins, PapD2 and CpxP, in the presence of CsgF[24].

We further investigated the role of Asn17 with a Congo red secretion assay (see Methods). The mutation of Asn17 to a similar hydrophilic tyrosine residue (N17Y) retained the curli formation, while mutation to a highly hydrophobic phenylalanine residue (N17F) nearly abolished curli formation (Fig. 2b). Considering that

CsgA is rich in hydrophilic residues (~50%), the mutated $CsgF_N$ pore having hydrophobic residues might not favor the transport of hydrophilic substrates. A mutation of Asn17 to a negatively charged aspartic acid residue (N17D) also reduced curli formation (Fig. 2b), probably owing to the repulsion by negatively charged residues (10 such residues) in CsgA.

Due to the limited pore size in $CsgF_N$, mutating Asn17 to a residue with a longer side chain may block the pore or reduce its diameter to obstruct the passage of the substrate. This was confirmed by the loss of the Congo red phenotype after mutating Asn17 to arginine (N17R) (Fig. 2b). Interestingly, transforming a N17R-mutated CsgF ($CsgF_{N17R}$) in a wild-type strain to create a competitive admixture with the wild-type CsgFG complex also largely reduced Congo red phenotype (Fig. 2c). This brought an idea to use $CsgF_{N17R}$ or part of $CsgF_{N17R}$ as an inhibitor to prevent substrate secretion. We split the N-terminal domain of $CsgF_{N17R}$ into two peptides, $PeptideN_{N17R}$ (residues 1–17) and $PeptideC_{N17R}$ (residues 17–36) (Fig. 2d and Supplementary Fig. 6d), and added them to the Congo red plates (Fig. 2e, f). Within the tested concentration gradient of 0.01–10 μM, $PeptideN_{N17R}$ markedly inhibited secretion, as judged by the reduced Congo red phenotype (Fig. 2e), while $PeptideC_{N17R}$ did not cause any significant visible inhibition (Fig. 2f). An ITC assay showed strong affinity of $PeptideN_{N17R}$ for CsgG (Supplementary Fig. 7a). In the presence of $PeptideN_{N17R}$, CsgF-CsgG complex formation was significantly decreased (Supplementary Fig. 7b). As $PeptideN_{N17R}$ is derived from $CsgF_N$, it should be able to competitively bind to the CsgG channel to interrupt the binding of original CsgF and admix with the wild-type CsgFG complex. Since CsgG channel β-barrel is directly exposed to the environment, such drugs do not have to pass through the bacterial outer membrane to effectivly inhibit curli formation.

**CsgG-CsgF-curli connection**. The CsgFG structure shows that $CsgF_N$ binds to CsgG. As CsgF can also bind to CsgB[25], it may act as a bridge to connect CsgG and curli (CsgA and CsgB) to assemble into an integrated complex of curli biogenesis system. $CsgF_C$ was flexible, and was observed as fuzzy densities on the extracellular side of the CsgFG channel (Fig. 1c and Supplementary Fig. 1d), which should be responsible for interacting with CsgB. To further clarify CsgF function, we first performed a pulldown assay using CsgG as a bait. CsgG can pulldown both the full-length CsgF and $CsgF_N$ but not $CsgF_C$ (Fig. 3a). Therefore, $CsgF_C$ is unlikely to interact with CsgG. We then used full-length CsgF, $CsgF_N$, and $CsgF_C$ to pulldown CsgB, separately. While $CsgF_N$ could not pulldown CsgB, full-length CsgF and $CsgF_C$ could (Fig. 3b), indicating that $CsgF_C$ is responsible for binding to CsgB. These results indicated that the N- and C-terminal domains of CsgF have distinct functions in binding to CsgG and CsgB, respectively. Congo red cell-association assays also confirmed that the curli-cell-associated phenotype was lost in the absence of $CsgF_N$ or $CsgF_C$ (Fig. 3c). To investigate CsgF and CsgB association with CsgA, we used CsgF and CsgB as baits to capture CsgA and found that CsgF did not directly interact with CsgA while CsgB did (Fig. 3d). Thus, CsgB was a connector to connect CsgF and CsgA. Altogether, the assembly of curli biogenesis system adopts a linear assembly mode of CsgG-$CsgF_N$-$CsgF_C$-CsgB-CsgA (Fig. 3e).

**Structural basis of CsgA recognition by CsgG**. It has been reported that $CsgA_{N22}$ can be co-immunoprecipitated with CsgG[21]. However, the exact mechanism of $CsgA_{N22}$ binding with CsgG channel is still unclear. To address this question, we co-expressed and purified $CsgA_{N22}$ (with a C terminal fusion protein PhoA and strep-tag II) with the CsgFG complex. The

size-exclusion chromatography and SDS-PAGE showed that $CsgA_{N22}$ can form a stable and well-behaved complex with CsgFG (Fig. 4a, b). Furthermore, deletion of $CsgA_{N22}$ results in the loss of Congo red phenotype (Fig. 4c), confirming that $CsgA_{N22}$ is indispensable for CsgA secretion.

To investigate how $CsgA_{N22}$ binds to the CsgFG complex, we determined the cryoEM structure of the CsgFG-$CsgA_{N22}$ complex with an imposed C9 symmetry at 3.34 Å resolution (Supplementary Fig. 8a–c). The overall architecture of CsgFG-$CsgA_{N22}$ complex was nearly identical to that of CsgFG complex (Supplementary Fig. 8d). The differences of CsgFG-$CsgA_{N22}$ complex from CsgFG complex include two additional densities on each CsgG subunit outside the CsgG periplasmic chamber (Supplementary Fig. 9). One being a part of $CsgA_{N22}$ (residues 2–7, sequence VVPQYG, Supplementary Fig. 9b) and the other being a CsgG loop (residues 104–110, Supplementary Fig. 9b) missing in the CsgFG complex in the absence of $CsgA_{N22}$. This loop was found to be flexible in CsgFG complex and stabilized after binding to $CsgA_{N22}$. From the CsgFG-$CsgA_{N22}$ complex structure and Congo red phenotype of $CsgA_{N22}$ deletion mutant, we concluded that CsgG not only function as a channel to transport CsgA substrate, but also as a recognition protein to specifically recognize CsgA at the initial stage of transport.

**Mechanism of CsgA recognition by CsgG**. Six $CsgA_{N22}$ residues ($V_2V_3P_4Q_5Y_6G_7$) were trapped in a crevice on CsgG surface mainly by hydrophobic interactions and hydrogen-bonds (Fig. 5a and Supplementary Fig. 10b). We further co-purified the CsgA N-terminus ($CsgA_{N6}$, residues $G_1V_2V_3P_4Q_5Y_6$,) with CsgG. Note that G1 was added to connect the signal peptide, and G7 was not included in the construct. $CsgA_{N6}$ showed similar ability to bind to CsgG as that by $CsgA_{N22}$ (Fig. 5b). ITC assays revealed the dissociation constant (Kd) for $CsgA_{N6}$ peptide binding with CsgG channel in vitro to be 23.8 μM (Supplementary Fig. 10c). Empirically, this binding affinity was neither strong nor weak, which might be favored by the dynamic binding and dissociation between CsgG and CsgA during secretion.

Using multiple sequence alignment (MSA), we found that the Val3 residue of CsgA was conserved and interacted with a small hydrophobic pocket surrounded by residues I121, L172, L236, and W237 of CsgG (Fig. 5a, c and Supplementary Fig. 10b). Congo red phenotype was lost by V3D mutation of CsgA (Fig. 5d), indicating that it prevents CsgA secretion. The Congo red phenotype was also lost by mutating the Ile121 or Trp237 residues of CsgG to a charged residue (Fig. 5e). Mutating Leu172 or Leu236 residues in CsgG to a charged residue inhibited curli formation less than I121D and W237D mutations (Fig. 5e), which implied relatively weak interactions between them (Leu172 and Leu236) and Val3 of CsgA, probably due to their relatively large distance from Val3. Gln5 residue is highly conserved as shown by MSA (Fig. 5c) and forms hydrogen-bonds with the main-chain residues Asp238 and Gly32 residues and side chain of Thr31 of CsgG (Fig. 5a). Congo red phenotype was also lost by Q5A mutation of CsgA (Fig. 5d), i.e., indicating that it prevents CsgA secretion. Similarly, Congo red phenotype was nearly lost by deleting single residue surrounding Gln5, including Asp238 and Gly32 or their neighboring residues (Trp237 and Thr31) of CsgG (Fig. 5e). The sixth residue of CsgA, Tyr6, was a relatively conserved aromatic residue as seen in MSA (Fig. 5c). Tyr6 interacts with the CsgG subunit to stabilize the adjacent CsgG loop region (residues 107–110, Fig. 5a). The Y6A mutation of CsgA reduced Congo red phenotype (Fig. 5d). Accordingly, CsgG loop deletion also largely reduced Congo red phenotype (Fig. 5e). Altogether, we found that CsgG recognizes CsgA mainly by binding to the conserved residues Val3, Gln5, and Tyr6.

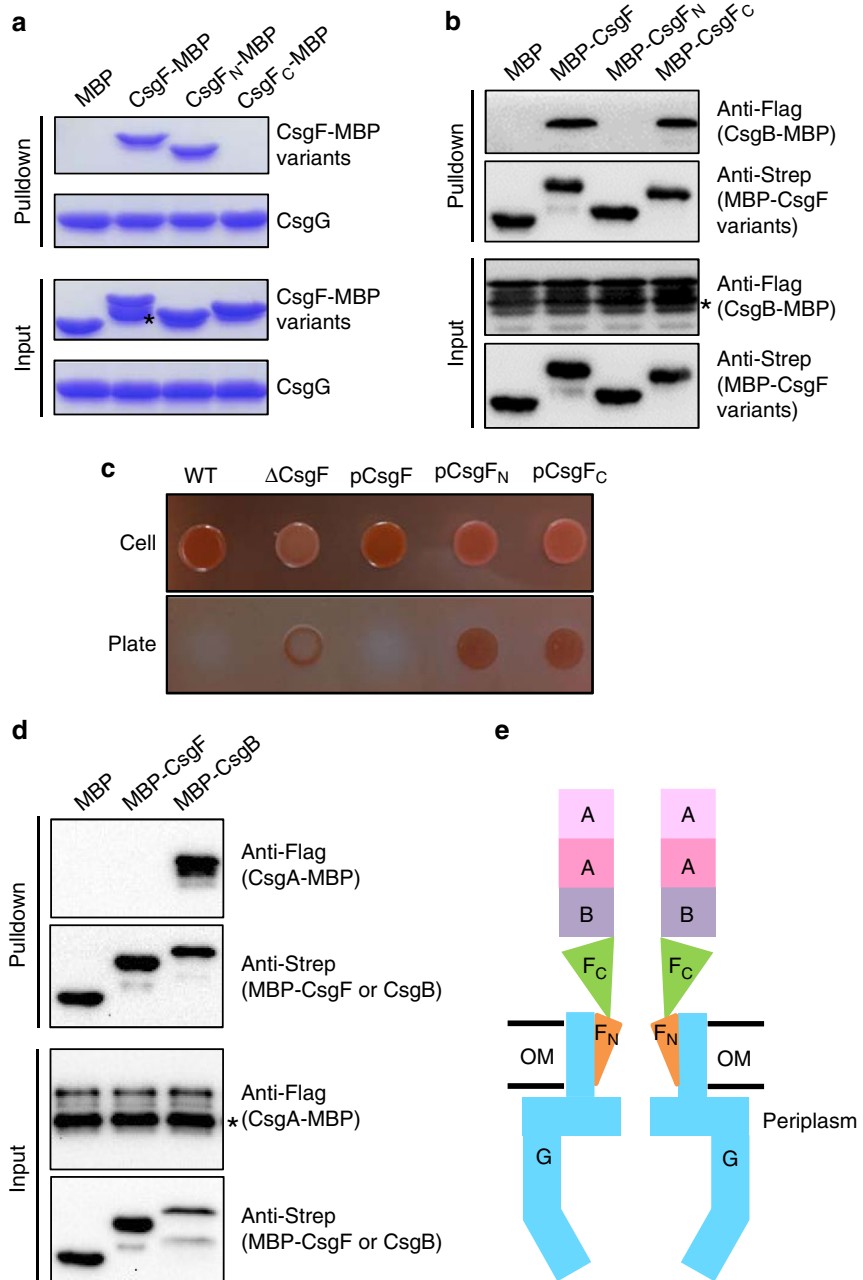

**Fig. 3 Interactions between major components of curli biogenesis system and its assembly model. a** Pulldown assay with CsgG and different parts of CsgF (C-terminal MBP tag). * Indicates CsgF degradation. Source data are provided as a Source Data file. **b** Pulldown assay with CsgB(C-terminal MBP tag) and different parts of CsgF (N-terminal MBP tag). * Indicates CsgB degradation. Source data are provided as a Source Data file. **c** Congo red cell-association assay for different parts of CsgF. Source data are provided as a Source Data file. **d** Pulldown assay of CsgF(N-terminal MBP tag), CsgB(N-terminal MBP tag) and CsgA(C-terminal MBP tag). * Indicates CsgA degradation. Source data are provided as a Source Data file. **e** Overall assembly model of curli biogenesis system. OM outer membrane.

**The role of CsgE in CsgA secretion.** Since the N-terminus of CsgA (N22 sequence) is used for substrate recognition and full-length CsgA can interact with CsgE[11], it is likely that the other parts of CsgA, five C-terminal repeating units (CsgA$_{R1-5}$), are involved in the interaction with CsgE. We performed a pulldown assay and confirmed that CsgE bound to CsgA$_{R1-5}$ responsible for amyloid fiber formation (Supplementary Fig. 11a). CsgE has also been reported to inhibit self-assembly of CsgA into amyloid fibers[24], which might be due to CsgE-CsgA$_{R1-5}$ interactions. Since CsgE also interacts with the CsgG channel[21,22] (Supplementary Fig. 11b and Supplementary Fig. 12a), we hypothesized that

CsgA, CsgE, and CsgG have a triangular relationship (Supplementary Fig. 11c).

We observed that CsgE deletion causes loss of Congo red phenotype in *E. coli* BW25113 (Supplementary Fig. 11d), indicating remarkably reduced curli secretion. However, Congo red phenotype was observed after CsgG over-expression in this CsgE-deleted strain (Supplementary Fig. 11d), reflecting the recovery of curli secretion. The similar phenomenon was also reported in a previous study with *E. coli* MC4100 strain, which also showed significant reduction and recovery of CsgA production due to CsgE deletion and CsgG over-expression,

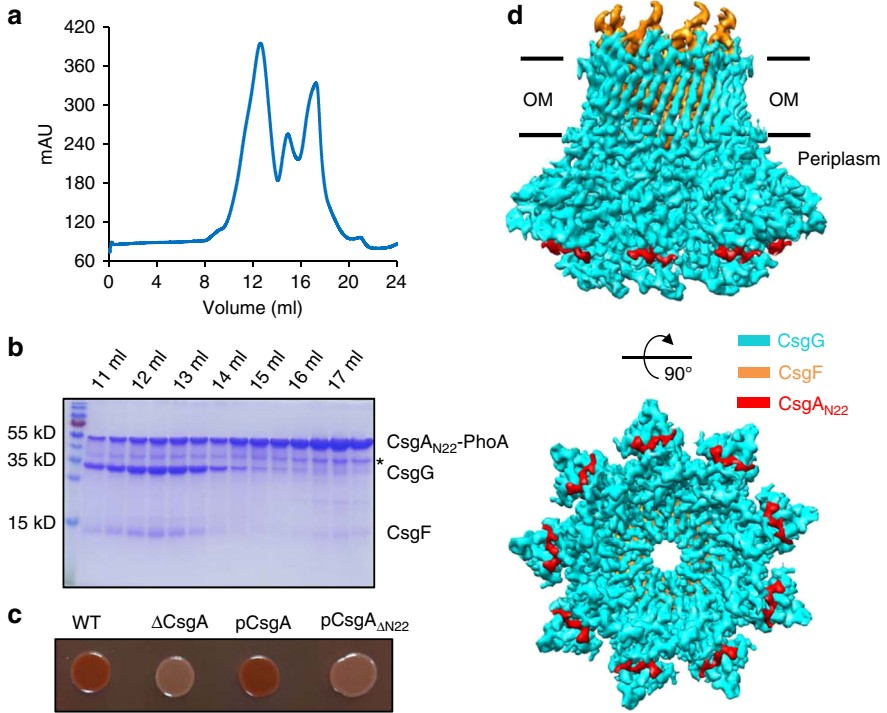

**Fig. 4 CsgA_N22 binding to CsgFG channel. a** Size-exclusion chromatogram of the purified CsgFG and CsgA_N22-PhoA complex. **b** SDS-PAGE analysis and Coomassie blue staining of the purified CsgFG and CsgA_N22-PhoA complex from SEC. * Indicates PhoA degradation. Source data are provided as a Source Data file. **c** Congo red secretion assay for CsgA_N22 truncation mutant. Source data are provided as a Source Data file. **d** Side (top) and top (bottom) view of the CsgFG-CsgA_N22 complex structure. CsgG, CsgF, and CsgA_N22 are in cyan, orange and red, respectively. OM outer membrane.

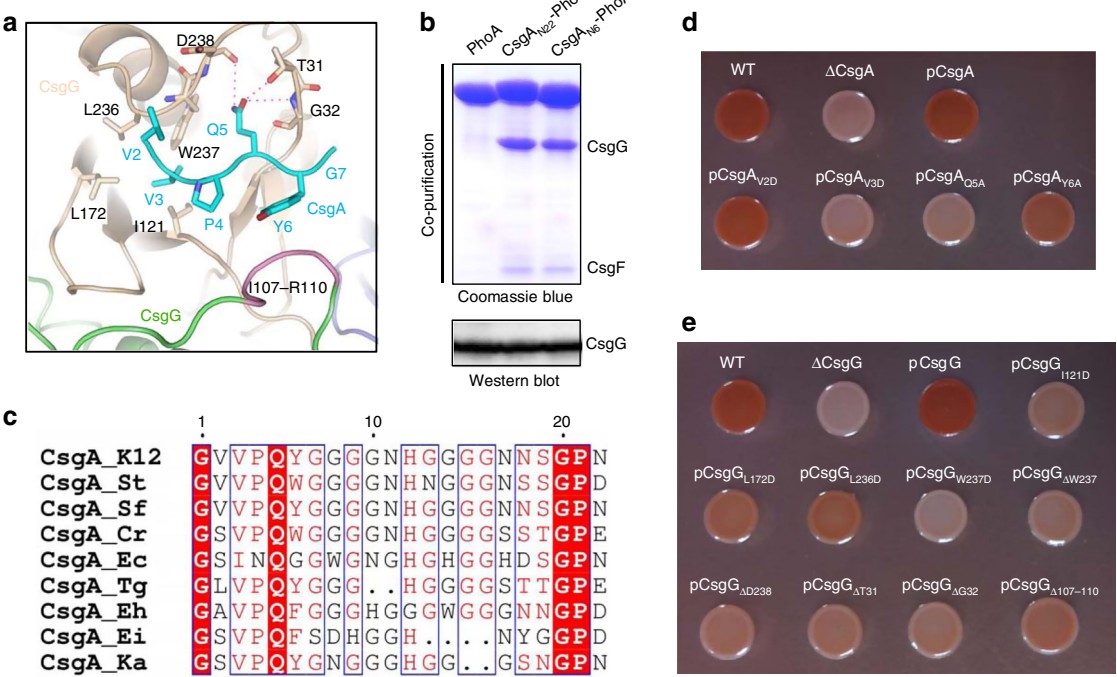

**Fig. 5 Analysis of interaction between CsgA_N22 and CsgG channel. a** Area of interaction between CsgA_N22 and the CsgFG channel. Six residues of CsgA_N22 are shown in cyan with side chains, while the two adjacent CsgG subunits are shown in wheat and green, respectively. Hydrogen bonds are shown as red dashed lines. **b** Co-purification assay of CsgA_N6 and CsgA_N22 with CsgG as detected by Coomassie blue staining. Detergent-solubilized CsgG supernatant was detected by western blotting. Source data are provided as a Source Data file. **c** Multiple sequence alignment of CsgA_N22 from several species. K12, *Escherichia coli* K12 strain; St, *Salmonella typhimurium* LT2 strain; Sf, *Shigella flexneri*; Cr, *Citrobacter rodentium* ICC168 strain; Ec, *Enterobacter cloacae* EcWSU1 strain; Tg, *Trabulsiella guamensis* ATCC 49490 strain; Eh, *Enterobacter hormaechei* ATCC 49162 strain; Ei, *Enterobacter intermedius*; and Ka, *Kosakonia sacchari* SP1 strain. **d** Congo red secretion assay for the single mutations in CsgA_N6. Source data are provided as a Source Data file. **e** Congo red secretion assay for the single mutations or deletions of CsgG corresponding to interactions with CsgA_N22. Source data are provided as a Source Data file.

respectively[24]. Based on these observations, it has been suggested that CsgE possibly enhances the efficiency of CsgG-mediated CsgA secretion[24]. CsgE deletion lowers translocation efficiency per single CsgG channel, thus reducing the amount of CsgA secreted, which further leads to loss of Congo red phenotype (White phenotype, Supplementary Fig. 11d, e). On the contrary, over-expression of CsgG protein in CsgE deletion strain increases the number of CsgG channels and compensates the low translocation efficiency per single channel. Consequently, the amount of CsgA secreted increases, leading to recovery of Congo red phenotype (Red phenotype, Supplementary Fig. 11d, e).

In the CsgFG-CsgA$_{N22}$ complex structure, we observed that CsgA$_{N6}$ was trapped in the CsgG recognition crevice outside CsgG periplasmic chamber and separated by the CsgG channel wall (Supplementary Fig. 12b). Therefore, the trapped CsgA$_{N22}$ cannot access the CsgG periplasmic chamber. Instead the other end of CsgA, CsgA$_{R1-5}$, may first enter the CsgG periplasmic chamber, and is then translocated through the CsgG pore (Supplementary Fig. 12b). As mentioned above, CsgE also interacts with CsgA$_{R1-5}$ and CsgG. Previously reported CsgE-CsgG complex structure shows that nine CsgEs form a cap-like structure at the entrance of the periplasmic chamber[22] (Supplementary Fig. 12a). Therefore, CsgE is likely to capture and confine CsgA$_{R1-5}$ into the CsgG periplasmic chamber to promote CsgA outward translocation (Supplementary Fig. 12b). In the absence of CsgE, CsgA$_{R1-5}$ may be able to enter the CsgG periplasmic chamber by molecular diffusion with a low efficiency.

## Discussion

We studied the curli biogenesis system using single-particle cryoEM and biochemical and functional analyses and uncovered the assembly curli biogenesis system, which plays a key role in understanding the secretion pathway. The CsgFG complex acts as a core for this assembly. CsgF uses its N terminal domain to connect with CsgG, and C terminal domain to associate with curli fibrils consisting of CsgA-CsgB. CsgF C-terminal domain was found to specifically interact with CsgB rather than CsgA. This assembly answers the questions about the association of curli fibrils with the cell. Considering 9:9 ratio of CsgF to CsgG, it is most likely that curli fibers also connect to CsgFG channel in the same ratio.

On the periplasmic side of the CsgFG channel, nine crevices on CsgG channel were found to be capable of capturing CsgA N-termini, which have been identified as recognition sites for CsgA. This finding suggests that CsgG not only involved in substrate transport, but also in substrate recognition. The six N-terminal residues of CsgA, CsgA$_{N6}$, were identified to be responsible for substrate recognition. Several residues in both CsgA$_{N6}$ and the CsgG crevices are conserved, which implies that mechanism for CsgA recognition is conserved among different species. Besides the CsgA$_{N6}$ recognition, The C-terminal domain of CsgA, CsgA$_{R1-5}$, was proved to be able to interact with CsgE. Since CsgA$_{N6}$ binds outside the CsgG periplasmic chamber and CsgE can form a cap-like structure at the bottom of this chamber[22], it is reasonable to propose that CsgA$_{R1-5}$ first enters this chamber under carrier of CsgE and then translocate through CsgFG channel. This may explain the significantly enhancement of CsgA translocation efficiency in the presence of CsgE.

The structural, biochemical, and functional results mentioned above imply a possible recognition-transportation model for CsgA secretion (Supplementary Fig. 13). After being translocated to the periplasmic space by the SecYEG machinery[27], CsgA is maintained as a soluble monomer by the periplasmic chaperon CsgC[18], and the CsgA N terminus (CsgA$_{N22}$) is then recognized and bound to the CsgG crevice, which may enrich CsgA near the

CsgG channel. After CsgG recognition, CsgE binds CsgA$_{R1-5}$, as well as CsgG, and carries CsgA$_{R1-5}$ to the CsgG periplasmic chamber. CsgA$_{R1-5}$ passes through the two pores located in the CsgFG secretion channel and is then secreted in the extracellular environment. Near the end of the secretion process, the CsgA N-terminus is released from CsgG and transported to the extra-cellular environment. Eventually, CsgA is nucleated by CsgB and forms fibrils connecting to CsgB-CsgF.

The discovery of the pore structure formed by CsgF$_N$ enabled us to design a peptide inhibitor to inhibit curli fiber formation. Unlike previous studies focusing on preventing CsgA fiber polymerization[12,28,29], we physically limited the CsgF$_N$ pore size to inhibit curli subunit secretion, and hence curli amyloid fiber formation. This should be a potentially powerful and efficient technique to discover new drugs for biofilm-related diseases. In summary, our structural, biochemical, and functional data provided the molecular basis for curli biogenesis system assembly, CsgA recognition and transport, as well as a feasible method for inhibiting the curli secretion.

## Methods

**Strain construction.** A one-step inactivation procedure was used to delete genes from the *E. coli* K12 BW25113(Cat: HAB(7-30)) strain[30]. Briefly, a kanamycin cassette was amplified from plasmid pKD4 using oligonucleotide pairs carrying 48-nucleotide extensions homologous to the regions adjacent to the gene to be deleted. After electroporation of 300 ng column-purified PCR product, kanamycin-resistant clones were selected and verified using colony-PCR. The kanamycin cassette was then excised using the plasmid pCP20. Gene deletions were confirmed using colony-PCR and sequencing.

**Plasmid construction.** The *CsgEFG* operon was amplified from the *E.coli* K12 DH5α(Cat: CB101-01) genome and cloned into the engineered pQlinkN plasmid containing a C-terminal strep-tag II using NdeI and BamHI restriction sites. For the CsgFG-CsgA$_{N22}$ complex, CsgEFG and CsgA$_{N22}$-PhoA were cloned to pQlinkN with C-terminal His-Flag tag and strep-tag II, respectively, and linked together by a ligation-independent cloning method[31]. The pQlinkN-BA vector was created by introducing the CsgBA promoter. All primers used in our study are presented in Supplementary Table 1.

**Protein expression and purification.** The *CsgEFG* and *CsgEFG/CsgA$_{N22}$-PhoA* constructs were transformed in the *E. coli* BL21 (DE3) (Cat: CB105-02)strain. The cells were grown at 37 °C in lysogeny broth (LB) and induced with 0.5 mM IPTG when the optical density of the culture at 600 nm reached ~1.2. After culturing for about 16 h at 18 °C, the cells were collected and resuspended with a buffer having 25 mM Tris-HCl at pH 8.0, 300 mM NaCl. The pellet was disrupted by emulsiflex-C3 (Avestin), and cell debris was removed by low-speed centrifugation for 10 min. After centrifuging the supernatant at 208400 g. (Beckman, 70Ti rotor) for 1 h, the membrane fraction was collected and solubilized in a buffer having 25 mM Tris-HCl at pH 8.0, 150 mM NaCl, 1% m/v n-Dodecyl-β-D-Maltopyranoside (DDM, Anatrace Company) and 1% m/v n-Dodecyl-N,N-Dimethylamine-N-Oxide (LDAO, Anatrace Company) for 1.5 h at 4 °C. The recombinant protein was purified using Strep-Tactin column (IBA Company), eluted with buffer containing 25 mM Tris-HCl at pH 8.0, 150 mM NaCl, 0.2% m/v LDAO and 2.5 mM des-thiobiotin (IBA), and further purified by size-exclusion chromatography (Superose 6 10/300 GL Increased, GE Healthcare Company). The peak fractions were analyzed by 15% SDS–PAGE and mass spectrometry. The sample was stored at −80 °C for future use.

**Cryo-electron microscopy.** A 4 μl drop of the sample (~0.6 mg/ml) was applied to a glow-discharged holey carbon grid (R2/2, 400 mesh, Quantifoil) with a home-made continuous carbon film (~5 nm thick), incubated for 15 s and blotted for 5 s at 13 °C and 100% humidity. The grid was then plunge-frozen in liquid ethane cooled by liquid nitrogen using FEI Vitrobot Mark IV (FEI Company). The grid was then transferred to an FEI Titan Krios electron microscope (FEI Company) operated at 300 kV. All images were recorded on a K2-Summit counting camera (Gatan Company) at a nominal magnification of 22,500 (yielding a 0.66 Å pixel size at super-resolution mode) using UCSFImage4[32]. All micrographs were dose-fractionated to 32 frames with a total exposure time of 8 s under a dose rate of ~8.2 counts per physical pixel/s. The total dose was ~50 e/Å². A defocus range of 1.3–2.5 μm was used for all micrographs.

**Image processing.** All micrographs were first processed by MotionCor2[33] with twofold binning (yielding a pixel size of 1.32 Å). Micrographs were then screened using dosef_logviewer (written by X.L.), and those with poor Thon rings were

removed. CTFFIND3 was used to estimate the defocus parameters[34]. Particle picking, 2D and 3D classification, 3D refinement and post-processing were all carried out using RELION[35].

For the CsgFG complex, 1,899 micrographs were used for particle picking, and the picked particles were extracted with a box size of 256 pixels. After two rounds of 2D classification, 372,680 particles were selected, subjected to 3D auto-refinement with C9 symmetry imposed, and initialized with a cylinder model generated by SPIDER[36], which resulted in a 3D reconstruction at 3.38 Å resolution with an auto-estimated B factor of −102.7 Å$^2$. For the CsgFG/CsgA$_{N22}$-PhoA complex, the procedure was similar to that of the CsgFG complex. A total of 139,702 particles were subjected to 3D auto-refinement with C9 symmetry and resulted in a 3D reconstruction at 3.34 Å resolution with an auto-estimated B factor of −173.9 Å$^2$ for final map sharpening.

**Model building**. For the CsgFG complex, the published CsgG crystal structure (pdb code: 4UV3)[22] was fitted into the cryo-EM map by Chimera[37] and manually adjusted in Coot[38] according to the density. The N terminus of CsgF was manually built de novo. Bulky residues (Phe, Tyr, and Arg) and unique sequence patterns were used as markers to guide and verify sequence assignment during the model building. The model was further refined in real space by PHENIX[39]. For the CsgFG/CsgA$_{N22}$-PhoA complex, the CsgFG model was fitted into the cryo-EM map by Chimera and the CsgG loop (T104-R110) was manually built in Coot. The CsgA$_{N22}$ peptide (VVPQYG) was also manually built in Coot. The model was further refined in real space by PHENIX. CryoEM data collection, refinement and validation statistics are presented in Supplementary Table 2.

**Circular dichroism**. CD spectra of CsgG and CsgF$_N$-MBP were measured using the Circular Dichroism Spectrometer Chirascan™ (Applied Photophysics Company). CsgG and CsgF$_N$-MBP were purified by size-exclusion chromatography in 8 mM Tris at pH 8.0, 50 mM NaCl, and 0.02% DDM before recording the CD spectra. For both individual proteins and the mixed complex (incubated in ice for about 3 h), the final concentrations of the CsgG nonamer and CsgF$_N$-MBP monomer were 0.5 μM and 4.5 μM, respectively. Spectra were recorded in 8 mM Tris at pH 8.0, 50 mM NaCl, and 0.02% DDM at 25 °C with a path length of 1 mm. Each spectrum represents the average of five scans, collected from 280 nm to 190 nm, with a spectral bandwidth of 1 nm and a response time of 0.5 s.

**ITC analysis**. The ITC assays were performed with the MicroCal ITC200 (GE Healthcare Company) at 25 °C. The DDM-solubilized CsgG or CsgFG were incubated with amphipole A8-35, and the detergent was removed by incubating with M2 beads overnight at 4 °C. The proteins were applied to gel filtration chromatography in buffer containing 10 mM HEPES at pH 7.5, and 50 mM NaCl before ITC analysis. The CsgF$_N$-MBP, CsgA$_{N6}$ peptide (G$_1$V$_2$V$_3$P$_4$Q$_5$Y$_6$) and PeptideN$_{N17R}$ (GTMTFQFRNPNFGGNPR) were analyzed in the same buffer as CsgG. For the CsgG/CsgF$_N$-MBP complex, nonameric CsgG (2 μM) was in the cell while monomeric CsgF$_N$-MBP (200 μM) was in the syringe. For the CsgG-CsgA$_{N6}$ complex, nonameric CsgG (10 μM) was in the cell while the CsgA$_{N6}$ peptide (1 mM) was in the syringe. For the CsgG-PeptideN complex, nonameric CsgG (2 μM) was in the cell while the PeptideN (60 μM) was in the syringe. The titration experiments consisted of an initial injection of 0.5 μL followed by 18 injections of 2 μL each at an interval of 150 s (for PeptideN) or 200 s (for CsgA$_{N6}$ peptide and CsgF$_N$-MBP) and a stirring rate of 750 rpm. The data were fitted using the Origin 7.0 software package of MicroCal ITC200 implementation, yielding the dissociation constants (Kd).

**Western blotting analysis**. Samples were added to SDS-loading buffer and boiled at 100 °C for 20 min. The boiled samples were separated on a 15% SDS-PAGE gel at 250 V for 30 min. The resolved proteins were transferred to Immobilon-P$^{SQ}$ transfer membrane (Merck Millipore Company) in ice for 80 min with a constant current of 330 mA. The transferred membrane was blocked in TBST buffer (25 mM Tris at pH 7.5, 150 mM NaCl, and 0.05% Tween 20) with 5% skimmed milk in 4 °C overnight. The transferred membrane was incubated with primary antibodies (anti-strep mouse monoclonal antibody with a dilution of 1:5500 [Sigma Company, Cat: SAB2702215]; anti-flag mouse monoclonal antibody with a dilution of 1:2750 [Beijing ComWin Biotech Company, Cat: CW0287M], anti-his mouse monoclonal antibody with a dilution of 1:3500 [EASYBIO Company, Cat: BE2019]) for 1 h at room temperature. After washing twice with TBST buffer, the transferred membrane was incubated with goat anti-mouse secondary antibody with a dilution of 1:6000 (Beijing ComWin Biotech Company, Cat: CW0102) for 1 h at room temperature. After further washing twice with TBST buffer and once with TBS buffer (25 mM Tris at pH 7.5, and 150 mM NaCl), the transferred membrane was detected with FluorChem FC3 (Protein Simple) by the enhanced chemiluminescence method.

**Pulldown assay**. Wild-type CsgG and CsgF variants were transformed to the *E. coli* BL21 (DE3) strain. The cells were induced with 0.5 mM IPTG for 16 h at 18 °C. The procedure of CsgG purification was similar to that of the CsgFG complex. After emulsiflex-C3 disruption and centrifugation, equivalent amounts of the CsgF variant supernatants were added to the detergent-solubilized wild-type CsgG

supernatant. After incubation for 30 min, the mixture was loaded into a Strep-Tactin column (IBA). The beads were washed four times with a buffer having 25 mM Tris-HCl at pH 8.0, and 150 mM NaCl, and eluted in 25 mM Tris-HCl at pH 8.0, 150 mM NaCl, and 2.5 mM desthiobiotin. Individual CsgG and CsgF variants were purified as input by Strep-Tactin and Ni-NTA, respectively. The samples were analyzed by 15% SDS–PAGE and stained with Coomassie blue.

The CsgA, CsgB and CsgF variants were transformed into *E. coli* BL21 (DE3) strain. The cells were induced with 0.5 mM IPTG for 4 h at 37 °C. After emulsiflex-C3 disruption and centrifugation, equivalent amounts of the CsgF variant supernatants were added to CsgB supernatant, or equivalent amounts of the CsgF and CsgA supernatants were added to the CsgA supernatant. After incubation for 20 min, the mixture was loaded to a Strep-Tactin column (IBA). The beads were washed four times with a buffer having 25 mM Tris-HCl at pH 8.0, 150 mM NaCl, and eluted in a buffer having 25 mM Tris-HCl at pH 8.0, 150 mM NaCl, and 2.5 mM desthiobiotin. Individual CsgA, CsgB and CsgF variants were purified as input by Ni-NTA and Strep-Tactin, respectively. The samples were analyzed by 15% SDS–PAGE and detected by western blotting.

The CsgE and CsgG were transformed in *E. coli* BL21 (DE3) strain. The cells were induced with 0.5 mM IPTG for 16 h at 18 °C. After emulsiflex-C3 disruption and centrifugation, equivalent amounts of the CsgE supernatant were added to detergent-solubilized wild-type CsgG supernatant. After incubation for 30 min, CsgE alone and CsgE-CsgG mixture were loaded to Strep-Tactin columns (IBA), respectively. The beads were washed for four times with a buffer having 25 mM Tris pH 8.0, 150 mM NaCl and 0.02% DDM and then eluted by an elution buffer (25 mM Tris pH 8.0, 150 mM NaCl, 0.02% DDM and 25 mM desthiobiotin). Individual CsgE and CsgG proteins were purified as input by Ni-NTA and Strep-Tactin, respectively. The samples were analyzed using 15% SDS–PAGE and detected using western blotting.

CsgF$_{GBD}$-MBP mutant variants were purified as soluble protein by Ni-NTA. CsgG were purified as membrane protein by Strep-Tactin and substituted by amphipole A8-35. CsgF$_{GBD}$-MBP mutant variants (36 μM) and CsgG nonamer (2 μM) were mixed together at 4 °C for 2 h and then incubated with equal volume of Strep-Tactin beads at 4 °C for 1 h. The protein-bound beads were washed for 4 times with washing buffer (25 mM Tris pH 8.0, 150 mM NaCl) and then eluted by an elution buffer (25 mM Tris pH 8.0, 150 mM NaCl and 2.5 mM desthiobiotin). The samples were analyzed by 15% SDS–PAGE and detected by western blotting.

**Congo red secretion and cell-association assays**. The *E. coli* K-12 strain BW25113 wild type (WT) and mutants were cultured overnight in LB at 37 °C. To induce curli production, 5 μl sample was spotted on YESCA plates (10 g/L casamino acids, 1 g/L yeast extract, and 20 g/L Bacto agar) supplemented with 100 μg/ml ampicillin and 90 μg/ml Congo red. The plates were incubated at 26 °C for 48 h. The colony was observed for dye binding after 48 h[22]. For overexpression assay in the ΔCsgE mutant strain, 3 μM IPTG was also supplemented in the YESCA plates. For the Congo red secretion assay, cells producing curli stained red, whereas mutants that failed to produce curli remained relatively white.

For the Congo red cell-association assay, after growing in YESCA plates for 48 h, the cells were scraped from the plates and the scraped plates were observed. The remaining plates with curli cell association appeared white, while the remaining plates without curli cell association appeared red.

**Co-purification assay**. Ligation-independent cloning linked constructs were transformed into the *E. coli* BL21 (DE3) strain. For each clone, the cells were grown at 37 °C in 3 L LB and induced with 0.5 mM IPTG for about 16 h at 18 °C. The cells were collected and disrupted by emulsiflex-C3 (Avestin). After low-speed and ultra-speed centrifugation, the membrane fraction was collected and solubilized in 25 mM Tris-HCl at pH 8.0, 150 mM NaCl, and 1.5% DDM for 1.5 h at room temperature. After ultra-speed centrifugation, 10 ml supernatant was loaded on Strep-Tactin column (IBA Company), eluted with buffer containing 25 mM Tris-HCl at pH 8.0, 150 mM NaCl, 0.02% DDM, and 2.5 mM desthiobiotin (IBA) and then analyzed by 15% SDS-PAGE. The detergent-solubilized supernatant was added to SDS-loading buffer and boiled at 100 °C for 20 min. The sample was then analyzed by western blotting.

**Competitive binding assay**. CsgF$_{GBD}$-MBP was purified as soluble protein by Ni-NTA. CsgG was purified as membrane protein by Strep-Tactin and substituted by amphipole A8-35. CsgF$_{GBD}$-MBP (18 μM) and PeptideN with different concentrations (0 μM, 1 μM, 3 μM, 9 μM, 27 μM and 81 μM) were mixed together at 4 °C for 1 h and then mixed with CsgG nonamer (2 μM) at 4 °C for 1 h. CsgF$_{GBD}$-MBP/PeptideN/CsgG mixtures were incubated with equal volume of Strep-Tactin beads at 4 °C for 1 h. The protein-bound beads were washed for four times with washing buffer (25 mM Tris pH 8.0, 150 mM NaCl) and then eluted by elution buffer (25 mM Tris pH 8.0, 150 mM NaCl, and 2.5 mM desthiobiotin). The samples were analyzed by 15% SDS–PAGE and detected by western blotting.

**Reporting summary**. Further information on research design is available in the Nature Research Reporting Summary linked to this article.

## Data availability

Data supporting the findings of this manuscript are available from the corresponding author upon reasonable request. A reporting summary for this Article is available as a Supplementary Information file.

The source data underlying Figs. 2b, c, e, f, 3a–d, 4b, c, 5b, d, e and Supplementary Figs. 1b, 3c, 5e, f, 7a, b, 10c, 11a, b, and d are provided as a Source Data file.

Density maps of the CsgFG and CsgFG-CsgA$_{N22}$ complexes are available through the EMDB with entry codes EMD-0841 and EMD-0842, respectively. Models of the CsgFG and CsgFG-CsgA$_{N22}$ complexes are deposited in the Protein Data Bank (PDB) with entry codes 6L7A and 6L7C, respectively.

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

## Acknowledgements

This work was supported by funds from the National Natural Science Foundation of China (31722015 and 31570730 to X.L.), the National Key Research and Development Program (2016YFA0501102 and 2016YFA0501902 to X.L.), Advanced Innovation Center for Structural Biology (to X.L.), Tsinghua-Peking Joint Center for Life Sciences (to X.L.) and One-Thousand Talent Program by the State Council of China (to X.L.). We thank Tsinghua University Branch of the China National Center for Protein Sciences Beijing for providing facility support in protein preparation, cryo-electron microscopy and computation.

## Author contributions

Z.Y., M.Y. and X.L. designed all experiments. Z.Y. and M.Y. performed all the experiments. J.C. performed part of the experiments. Y.Z., M.Y. and X. L. wrote the manuscript. All authors contributed to the data analysis and manuscript revision.

## Competing interests

The authors declare no competing interests.
