## [Peer Review File · Nature Communications]

Reviewers' Comments:

Reviewer #1:

Remarks to the Author:

Li and coworkers elucidate a high-resolution structure of the CsgFG secretion channel complex. The structure of the FG complex revealed a pore formed by CsgF that ultimately serves to anchor curli fibers to the cell. The authors also describe interactions between CsgA subunits and CsgG that guide secretion. Overall, the study provides a significant step forward in the understanding of the curli secretion process. Finally, the authors propose a model for curli subunit secretion. Most of my comments are relatively minor, but they might help the authors while revising their manuscript.

Comments to consider in a revised manuscript:

CsgE is not seen as part of the CsgGF complex even though CsgE is presumably co-expressed. Goyal et al. found that CsgE caps at the base of CsgG, and here the authors show that CsgG can pull-down CsgE-MBP (Extended data Fig 11b). However, none of the structure presented here demonstrates the presence of CsgE in the CsgGF complex. Is the interaction between CsgG and CsgE transient? It would be useful to provide the dissociation constant of CsgG and CsgE interaction.

It is not always clear where and when MBP versions of proteins were being used. At the very least this would allow the reader to know how or if the MBP tags might be interfering or changing protein-protein interactions.

The F7D variant of CsgF-MBP perturbs the interaction with CsgG (Extended data Fig 5e) but the Congo red assay (Extended data Fig 5f) suggests that curli formation is mostly unaffected. Are the levels of secreted and polymerized CsgA unaffected by the F7D CsgF variant?

As a general note, the authors use Congo red binding as a proxy for CsgA secretion in several figures. Although Congo red binding is often a useful first readout, it has limitations and is not a direct readout of curli subunit secretion. For example, the lack of Congo red binding in Fig 5d could indicate that CsgA variants such as V3D, Q5D fail to polymerize on the cell surface. A direct measure of secretion (or lack of secretion) is needed certainly needed.

Fig 4d shows that N22 of CsgA bind to CsgG. Are the authors suggesting that 9 CsgA molecules could simultaneously engage the CsgG nonamer complex?

According to the second and last step in the proposed model CsgE attaches/detaches to/from CsgG form for every CsgA entry. What is driving the dissociation of CsgE?

Although the pull-down assays show that CsgA R1-5 interacts with CsgE and that N22 of CsgA binds to CsgG, the authors need to justify how they arrived at the binding events in the proposed model. In general, most of the steps in the model are not directly supported by data and are therefore superfluous.

Minor comments

Gene names to be italicized. In general, it was hard to tell when the authors were referring to genes and proteins.

In Extended data Fig 5f, the image labels are not sufficient for the reader to know what is happening.

How the experiment was carried out for the Fig 2e and f?

In extended Fig 2b correct 61th residue is labeled as Pho correct it to Phe.

Line 182, change CSgA to CsgA.

Reviewer #2:

Remarks to the Author:

The main focus of this paper is the use of electron microscopy in the structure determination of the Curli pore CsgG, in complex with CsgF and the N-terminal leader sequence of CsgA.

This is an important and novel structure, carefully determined and their conclusions are backed up by well-designed biochemical and cellular assays. The structures provide genuine new insight into variety of aspects: i) architecture of the constructions within the Curli secretion pore, ii)

recruitment of substrate and iii) inhibition by blocking the opening. This paper is definitely a priority to publish and is suitable to Nature Comms. However, the following should be addressed:

- 1) The work is nice and carried out proficiently, but the English grammar within manuscript could be much better. The corresponding author should seek advice and a proofing from a native English speaker.
- 2) The authors could go further with the analysis of their structure and add some sensible modelling and simulation. For example: a structure of CsgF is known from solution NMR (pdb: 5M1U), so the C-terminus can be modelled onto their complex and discussed (also the models of CsgA could be used to augment their structure further). Finally, some single-point mutations in the C-terminus of CsgF that abrogate the CsgA interaction could be identified and this would complement the work.
- 3) Furthermore, can any additional residues in the N22 peptide be model/simulated. If so, are there any interesting contacts that can be probed experimentally.
- 4) The authors do not mention CsgC in their model in Fig 6 and discussion. For completeness this should be added.
- 5) Furthermore, the authors could comment further on the energy source/flow for secretion and how is the substrate-N22 released from the CsgG pore
- 6) In the peptide inhibitor study the authors use two peptides of CsgF ie. the N- and the C-terminal halves. Based on the structure the C-terminal helix would seem the most appropriate peptide. The manuscript would benefit from a few more designed peptides to refine this study further.

Reviewer #3:

Remarks to the Author:

The paper significantly builds on the earlier CsfG crystallographic structure by presenting the structure of the secretion channel complex CsgF(Nter 36 residues)-CsgG with and without curli substrate. The cryoEM analysis is at sufficient resolution to model side chains, and the structures presented are of apparent good quality. Based on this, the authors propose a mechanism for substrate recognition and a novel approach for inhibiting secretion by restricting the size of the pore –a cool idea, especially given that the pore is in the OM so easily accessible by potential drugs. The paper also presents mutagenesis, CongoRed assays, pull downs and ITC data to elucidate how components of this system potentially interact with each other. Beyond the need for significant grammatical editing, the data is clearly presented and supports the final model presented in Figure 6. Collectively the work is a significant advance that will be of interest to the broad secretion and anti-bacterial development community.

Revisions:

In the abstract and throughout the manuscript, the authors should clarify that they have characterized the complex structure of GsgG with the N-terminal 36 amino acids of CsgF and not the full 120 amino acid protein. Also for CsgA, it is the N-terminal 7 residues not the full protein. A designation of CsgF1-36 rather than CsgF etc would be appropriate. Also in the text or methods please stipulate regions not observed in all structures.

In the Introduction – a sentence explaining the breadth of bacterial species with the curli system would be informative (Gram positive, Gram negative, prominent pathogens etc).

Also a summary and references of what is known structurally about the system prior to this study including prior GsgG crystal structure etc.

Please provide standard deviations for the ITC assays, with replicate value (should be at least

n=3). A control experiment titrating MBP into CsgG would also be informative.

Line 79 Provide the resolution of the prior CsgG monomer and rmsd's.

Line 80 Extended Data Figure 3 – reword for clarity (“another crystal structure” implies a second crystal structure here). - again rmsd here would be beneficial, also labels for N-ter, C-ter, and some intervening landmark regions (loop on outer barrel etc)

Line 101 In Extended data Fig.5c, given their critical role for CsgFG complex formation, please show the residues of CsgG which interact with F5D, F7D and F12D of CsgF.

Line 167, In Extended Data Figure 8. Only a C9 map is provided. Please provide a C1 map and stats as well. Are there unsymmetrized features you may be missing by enforcing the C9?

Line 172 and Extended Data Figure 9. In discussing a newly resolved loop in CsgG with the CsgA bound structure: it would be much more informative to provide an actual model of this area to depict the new loop region rather than a surface map representation which is not clear at all (perhaps a ribbon with transparent surface and depth cueing). Is there significance to this newly resolved loop? Where is it in reference to the CsgA binding site?

Line 182 – This sentence is not clear – the construct design should be better clarified for the reader.

Line 242 Reference required for translocation of the system to the OM by SecYEG

Line 419-429 Reference required for the Congo red secretion assay/cell-association assay

In Extended data Fig.3b, Kd for CsgFN and CsgG is 237nM. In Extended data Fig.7A, Kd for PeptideNn17r and CsgG is 12.7nM. Can the authors explain why? Since PeptideCn17r also interacts with CsgG (Extended data Fig.5d), CsgFN should bind to CsgG more strongly in theory.

Outer membrane (OM) labelling of figure 1A is not properly aligned.

Every structure/model showing the protein in a membrane should have clear labels for the outside/periplasm – Figures 1a/2a/3e/4d/6, Extended Data Figure 6/11c/12

Extended Data Figure 10a – Tyrosine 6 in the figure – the side chain appears misplaced - should be localized to the density coming off of the C-beta? Having the model for the rest of the density surrounding would help clarify.

Reviewer #1 (Remarks to the Author):

Li and coworkers elucidate a high-resolution structure of the CsgFG secretion channel complex. The structure of the FG complex revealed a pore formed by CsgF that ultimately serves to anchor curli fibers to the cell. The authors also describe interactions between CsgA subunits and CsgG that guide secretion. Overall, the study provides a significant step forward in the understanding of the curli secretion process. Finally, the authors propose a model for curli subunit secretion. Most of my comments are relatively minor, but they might help the authors while revising their manuscript.

A: Thanks for the constructive suggestions to improve our manuscript.

Comments to consider in a revised manuscript:

1. CsgE is not seen as part of the CsgGF complex even though CsgE is presumably co-expressed. Goyal et al. found that CsgE caps at the base of CsgG, and here the authors show that CsgG can pull-down CsgE-MBP (Extended data Fig 11b). However, none of the structure presented here demonstrates the presence of CsgE in the CsgGF complex. Is the interaction between CsgG and CsgE transient? It would be useful to provide the dissociation constant of CsgG and CsgE interaction.

A: Thanks for the suggestion. We have tried many times to purified CsgE-CsgFG complex according to the method described by Goyal et al in order to obtain the structure of CsgE-CsgFG complex. However we never got stable CsgE-CsgFG complex, while CsgG could pull-down CsgE (detected by western blot). As suggested by the reviewer, we performed ITC experiment and just measured very weak interaction in mM level (see the figure below). We also think that the transient interaction between CsgE and CsgG is possible.

2. It is not always clear where and when MBP versions of proteins were being used. At the very least this would allow the reader to know or if the MBP tags might be interfering

or changing protein-protein interactions.

A: Thanks for the suggestion. Ser-Ala (SA) linkers were used between all the MBP and target proteins to avoid the influence of MBP as more as possible. To avoid confusion, we added clearer descriptions in the related figure legends about where and when MBP versions of proteins were used.

3. The F7D variant of CsgF-MBP perturbs the interaction with CsgG (Extended data Fig 5e) but the Congo red assay (Extended data Fig 5f) suggests that curli formation is mostly unaffected. Are the levels of secreted and polymerized CsgA unaffected by the F7D CsgF variant?

A: Thanks for the suggestion. We also observed the different experimental results of F7D CsgF variants in the Congo red assay and pulldown assay. We have repeated the two experiments for several times and always got identical results. As the scraped plate (Congo red cell-association assay) of F7D CsgF variant appears faint red, there might exist weak interaction between F7D CsgF variant and CsgG, which might make the secretion mostly unaffected. And the weak interaction does weaken the connection between CsgAB fibers and the cell through CsgF. Because the interaction might be too weak and beyond the detection limit of western blot, we didn't see binding in Extended data Fig 5e.

4. As a general note, the authors use Congo red binding as a proxy for CsgA secretion in several figures. Although Congo red binding is often a useful first readout, it has limitations and is not a direct readout of curli subunit secretion. For example, the lack of Congo red binding in Fig 5d could indicate that CsgA variants such as V3D, Q5D fail to polymerize on the cell surface. A direct measure of secretion (or lack of secretion) is certainly needed.

A: Thanks for the suggestion. I am sorry that we can't perform western blot to detect the level of CsgA since we cannot find available CsgA antibody. We agree with the opinion of the reviewer that Congo red binding, strictly speaking, is not a direct readout of curli subunit secretion since lack of phenotype of Congo red may result from lack of CsgA secretion or failure of polymerization of CsgA.

However, except for fig.4c and fig5d, we didn't change sequence of the CsgA and focused on the change of CsgG, as well as N terminus of CsgF or CsgE, which should not cause the failure of polymerization of CsgA, but we observed their influences to the CsgA secretion. Although these are not direct evidences, the Congo red assay is still enough to evaluate the secretion for those experiments with wild-type CsgA.

As for fig.4c and fig5d, we only changed the targeting sequence of CsgA (N22 sequence), and didn't change the R1-R5 core repeats which are responsible for the polymerization of CsgA. Since the N22 sequence of CsgA is not the core of CsgA fibers (Chapman, M.R. *et al. Science*, 2002), we think the change of N22 sequence of CsgA should not affect the polymerization of CsgA.

5. Fig 4d shows that N22 of CsgA bind to CsgG. Are the authors suggesting that 9 CsgA molecules could simultaneously engage the CsgG nonamer complex?

A: We think that it is possible that 9 CsgAs bind to all 9 pockets. The densities in the pockets are strong, indicating high occupancy of CsgA N-terminus in the pockets. Further, we tried to perform cryoEM 3D alignment and reconstruction of CsgA_{N22}-CsgFG complex (see the figure below) without imposing symmetry, and all pockets also showed strong densities of CsgA N-terminus. Anyway, our experiments only suggest the binding of CsgA N-terminus in these pockets with high occupancy. It is hard to draw a conclusion about how many full-length CsgA binding on one CsgG simultaneously under physiological condition.

6. According to the second and last step in the proposed model CsgE attaches/detaches to/from CsgG form for every CsgA entry. What is driving the dissociation of CsgE? Although the pull-down assays show that CsgA R1-5 interacts with CsgE and that N22 of CsgA binds to CsgG, the authors need to justify how they arrived at the binding events in the proposed model. In general, most of the steps in the model are not directly supported by data and are therefore superfluous.

A: Thanks for the suggestion. On the basis of the ITC result, the binding is weak between CsgE and CsgG. Hence, we speculate that the molecule dynamics drive the dissociation of CsgE from CsgG. We agree that the proposed model without more experimental supporting data is weak. So we move the model into the extended figure, and just raise and discuss the model as our hypothesis in the discussion section.

Minor comments

7. Gene names to be italicized. In general, it was hard to tell when the authors were referring to genes and proteins.

A: Thanks for the suggestion. We have checked and corrected the gene name in the revised manuscript.

8. In Extended data Fig 5f, the image labels are not sufficient for the reader to know what is happening.

A: Thanks for the suggestion. We have added the name of images into the figure in the

revised manuscript.

9. How the experiment was carried out for the Fig 2e and f?

A: We dissolved the PeptideN17R and PeptideC17R, respectively, with different concentrations into the Congo Red plates, and spotted the *E.coli* onto the plates. After 48h, we took photos for these plates.

10. In extended Fig 2b correct 61th residue is labeled as Pho correct it to Phe.

Line 182, change CSgA to CsgA.

A: Thanks for the suggestion. We have corrected them in the revised manuscript.

Reviewer #2 (Remarks to the Author):

The main focus of this paper is the use of electron microscopy in the structure determination of the Curli pore CsgG, in complex with CsgF and the N-terminal leader sequence of CsgA.

This is an important and novel structure, carefully determined and their conclusions are backed up by well-designed biochemical and cellular assays. The structures provide genuine new insight into variety of aspects: i) architecture of the constructions within the Curli secretion pore, ii) recruitment of substrate and iii) inhibition by blocking the opening. This paper is definitely a priority to publish and is suitable to Nature Comms. However, the following should be addressed:

A: Thanks for the constructive suggestions to improve our manuscript.

1) The work is nice and carried out proficiently, but the English grammar within manuscript could be much better. The corresponding author should seek advice and a proofing from a native English speaker.

A: Thanks for the suggestion. Sorry for the grammatical problems, we used a language service to improve the revised manuscript.

2) The authors could go further with the analysis of their structure and add some sensible modelling and simulation. For example: a structure of CsgF is known from solution NMR (pdb: 5M1U), so the C-terminus can be modelled onto their complex and discussed (also the models of CsgA could be used to augment their structure further). Finally, some single-point mutations in the C-terminus of CsgF that abrogate the CsgA interaction could be identified and this would complement the work.

A: Thanks for the suggestion. Unfortunately, the densities of CsgF C-terminal domain are very weak due to their high flexibility, which are only visible with low contour threshold. As suggested by the reviewers, we have tried to fit the CsgF C-terminal NMR model into our map (see the figure below, panel a is a side view of the model and our map, panel b is a locally enlarged CsgF C-terminal domain, and panel c is a top view). Because the low quality of the density map in this region, there is no guarantee that the model fitting is correct, accordingly, it is hard to find some meaningful single-point mutations. Therefore, we won't use this result in this work. We are now working this

region with consideration of structural flexibility by cryoEM, and expecting to get its high-resolution structure in the future.

3) Furthermore, can any additional residues in the N22 peptide be model/simulated. If so, are there any interesting contacts that can be probed experimentally.

A: Thanks for the suggestion. We repeated the pulldown experiments in figure 5b, and performed one more measurement for the interaction between the additional residues in CsgA_{N22} and CsgG (see 4th column in the figure below). We carried out co-purification assay of CsgA_{N22}, CsgA_{N6}, CsgA_{N7-22} with CsgG detected by Coomassie blue staining. Before pulldown, CsgG was detected as control in the detergent-solubilized supernatant by western blotting. From the result of pulldown assay, we didn't see interaction between the CsgA_{N7-22} and CsgG. Therefore, there is only very weak or even no interaction between the CsgA_{N7-22} and CsgG.

4) The authors do not mention CsgC in their model in Fig 6 and discussion. For completeness this should be added.

A: Thanks for the suggestion. We have added some discussion about CsgC in the revised manuscript and figure legend. Now the model has been moved to Extended Data Fig. 13.

5) Furthermore, the authors could comment further on the energy source/flow for secretion and how is the substrate-N22 released from the CsgG pore

A: Thanks for the suggestion. According to the work by Goyal, P. *et al.*, since there are no proton gradients at outer membrane, we also think that the entropic potential generated by the concentration difference of CsgA between the periplasm and extracellular environment drives the secretion. And the substrate-N22 move outwards via Brownian diffusion.

6) In the peptide inhibitor study the authors use two peptides of CsgF ie. the N- and the C-terminal halves. Based on the structure the C-central helix would seem the most appropriate peptide. The manuscript would benefit from a few more designed peptides to refine this study further.

A: Thanks for the suggestion. We are now working on some shorter peptides, and considering small molecular inhibitors of CsgG in the future work. Until now, the N-terminal peptide with 17 residues is still the best one.

Reviewer #3 (Remarks to the Author):

The paper significantly builds on the earlier CsfG crystallographic structure by presenting the structure of the secretion channel complex CsgF(Nter 36 residues)-CsgG with and without curli substrate. The cryoEM analysis is at sufficient resolution to model side chains, and the structures presented of apparent good quality. Based on this, the authors propose a mechanism for substrate recognition and a novel approach for inhibiting secretion by restricting the size of the pore –a cool idea, especially given that the pore is in the OM so easily accessible by potential drugs. The paper also presents mutagenesis, CongoRed assays, pull downs and ITC data to elucidate how components of this system potentially interact with each other. Beyond the need for significant grammatical editing, the data is clearly presented and supports the final model presented in Figure 6. Collectively the work is a significant advance that will be of interest to the broad secretion and anti-bacterial development community.

A: Thanks for the constructive suggestions to improve our manuscript. And sorry for the language problem, we had improved the language using a language editing service.

Revisions:

1. In the abstract and throughout the manuscript, the authors should clarify that they have characterized the complex structure of GsgG with the N-terminal 36 amino acids of CsgF and not the full 120 amino acid protein. Also for CsgA, it is the N-terminal 7 residues not the full protein. A designation of CsgF1-36 rather than CsgF etc would be appropriate. Also in the text or methods please stipulate regions not observed in all structures.

A: Thanks for the suggestion. In our work to get the cyroEM map of the CsgFG complex, we used full-length CsgF rather than just CsgF N-terminal 36 residues. While we only got high resolution at the CsgF N-terminal, we still could see some low-resolution densities of the CsgF C-terminal (see the figure below, we show the map with densities of CsgF C-terminal with a trial of fitting corresponding NMR structure in cyan color into it). Similar for CsgA_{N22}, we used the CsgA N-terminal 22 residues but only 6 residues were visible and determined at high resolution. Therefore, we think it is reasonable to use the name of CsgF and CsgA_{N22}, rather than just the parts with high resolution. We had checked through the entire manuscript to make sure that our description for name and region was clear.

2. In the Introduction – a sentence explaining the breadth of bacterial species with the curli system would be informative (Gram positive, Gram negative, prominent pathogens etc).

A: Thanks for the suggestion. We have explained it in this sentence “The curli biogenesis system in *Enterobacteriaceae* such as *Salmonella* species and *Escherichia coli* (*E.coli*), also known as the nucleation-precipitation secretion system or type VIII secretion system, is responsible for secreting curli subunits into the extracellular milieu to form cell associated nonbranching and highly aggregative amyloid fibrils^{2,3}.” in the revised manuscript.

3. Also a summary and references of what is known structurally about the system prior to this study including prior GsgG crystal structure etc.

A: Thanks for the suggestion. We have added more details about the known structures in the introduction section in the revised manuscript.

4. Please provide standard deviations for the ITC assays, with replicate value (should be at least n=3). A control experiment titrating MBP into CsgG would also be informative.

A: Thanks for the suggestion. We have provided it on the Extended Data Fig. 3, Extended Data Fig. 7 and Extended Data Fig. 10 in the revised manuscript, respectively.

5. Line 79 Provide the resolution of the prior CsgG monomer and rmsd's.

A: Thanks for the suggestion. The resolution of prior CsgG nonamer is 3.59 Å. The rmsd between the CsgG in our CsgGF complex and prior CsgG nonamer is 0.742 Å. We have added these information in the revised manuscript.

6. Line 80 Extended Data Figure 3 – reword for clarity (“another crystal structure” implies a second crystal structure here). - again rmsd here would be beneficial, also labels for N-ter, C-ter, and some intervening landmark regions (loop on outer barrel etc)

A: Thanks for the suggestion. We have revised the corresponding sentence to make it clearer, and added rmsd value in the revised manuscript. A panel with single subunits from two structures is added into this figure (Extended data Fig. S3b in the revised manuscript) in order to show the comparison of landmark regions.

7. Line 101 In Extended data Fig.5c, given their critical role for CsgFG complex formation, please show the residues of CsgG which interact with F5D, F7D and F12D of CsgF.

A: Thanks for the suggestion. We have showed the hydrophobic interaction between the F5D, F7D and F12D of CsgF and the surrounding CsgG in the revised manuscript.

8. Line 167, In Extended Data Figure 8. Only a C9 map is provided. Please provide a C1 map and stats as well. Are there unsymmetrized features you may be missing by enforcing the C9?

A: Thanks for the suggestion. We have provided the density map with C1 symmetry (Figure below). Panel a show the map of CsgFG_CsgA_{N22} in side view and top view. Panel b is the structural comparison of CsgFG_CsgA_{N22} in C1 and C9 symmetry. Panel c is the structural comparison of CsgFG_CsgA_{N22} in C1 symmetry and CsgFG in C9 symmetry. By comparison these maps, we don't see new features in the map without imposed symmetry.

9. Line 172 and Extended Data Figure 9. In discussing a newly resolved loop in CsgG with the CsgA bound structure: it would be much more informative to provide an actual model of this area to depict the new loop region rather than a surface map representation which is not clear at all (perhaps a ribbon with transparent surface and depth cueing). Is there significance to this newly resolved loop? Where is it in

reference to the CsgA binding site?

A: Thanks for the suggestion. We provided an actual model of the newly resolved loop in the revised manuscript, and optimized its representation to be clear (see figure below). And this newly resolved loop (purple model) is closed to the CsgA N-terminus (orange model), thus interacts with the CsgA N-terminus, and is stabilized during the CsgA N-terminus binding. The loop deletion can also largely reduce the Congo red phenotype (see Fig. 5e).

10. Line 182 – This sentence is not clear – the construct design should be better clarified for the reader.

A: Thanks for the suggestion. We have revised sentence to make construct design clearer in the revised manuscript.

11. Line 242 Reference required for translocation of the system to the OM by SecYEG

A: Thanks for the suggestion. The reference has been added in the revised manuscript.

12. Line 419-429 Reference required for the Congo red secretion assay/cell-association assay

A: Thanks for the suggestion. The reference has been added in the revised manuscript.

13. In Extended data Fig.3b, K_d for CsgFN and CsgG is 237nM. In Extended data Fig.7A, K_d for PeptideNn17r and CsgG is 12.7nM. Can the authors explain why? Since PeptideCn17r also interacts with CsgG (Extended data Fig.5d), CsgFN should bind to CsgG more strongly in theory.

A: Thanks for the suggestion. We have repeated the ITC experiments for several times, and always got similar results. We also don't have good explanation for this difference. A possibility is that the large CsgF protein may have some points with negative influence to the binding and thus reduce the affinity, and the much smaller peptides are easier to go in to the CsgG channel and bind on it.

14. Outer membrane (OM) labelling of figure 1A is not properly aligned.

A: Thanks for pointing out this. We have fixed it in the revised manuscript.

15. Every structure/model showing the protein in a membrane should have clear labels for the outside/periplasm – Figures 1a/2a/3e/4d/6, Extended Data Figure 6/11c/12

A: Thanks for the suggestion. We have revised all these figures in the revised manuscript.

16. Extended Data Figure 10a – Tyrosine 6 in the figure – the side chain appears misplaced - should be localized to the density coming off of the C-beta? Having the model for the rest of the density surrounding would help clarify.

A: Thanks for the suggestion. We have checked and verified the density of Tyrosine 6, and it is placed correctly. We make a figure to improve the representation of CsgA_{N6} and the adjacent loop in CsgG (see figure below, which has been updated to Extended data Fig. S10a in the revised manuscript). The green model is CsgA_{N6}, and the orange model is the loop and several residues in CsgG.

Reviewers' Comments:

Reviewer #1:

Remarks to the Author:

The authors have adequately addressed most of my comments. The only outstanding comment is regarding the use of Congo Red binding as a sole indicator of CsgA secretion. Although I agree with the authors that the mutations made are likely affecting CsgA secretion (and not CsgA polymerization into a Congo red-binding fiber), it is not definitive. For example, if a mutant variant in CsgG affected the secretion or positioning of CsgB or CsgF but NOT CsgA then the phenotype would be interpreted incorrectly by Congo red assays alone.

Ideally, an independent and direct measure of CsgA secretion would go a long way to solidifying any conclusions made.

Reviewer #2:

Remarks to the Author:

Responses are satisfactory

Reviewer #3:

Remarks to the Author:

The authors have addressed the concerns of the initial review adequately.